# Statistical Analysis of the Pouring Method’s Influence on the Distribution of Metallic Macrofibres into Vibrated Concrete

**DOI:** 10.3390/ma16041404

**Published:** 2023-02-07

**Authors:** Laura Gonzalez, Jose Sainz-Aja, Álvaro Gaute Alonso, Jokin Rico, Albert de la Fuente Antequera, Ignacio Segura, Carlos Thomas

**Affiliations:** 1INGECID S.L. (Ingeniería de la Construcción, Investigación y Desarrollo de Proyectos), E.T.S. de Ingenieros de Caminos, Canales y Puertos, Av./Los Castros 44, 39005 Santander, Spain; 2LADICIM (Laboratory of Materials Science and Engineering), University of Cantabria. E.T.S. de Ingenieros de Caminos, Canales y Puertos, Av./Los Castros 44, 39005 Santander, Spain; 3GiaDe (Grupo de Instrumentación y Análisis Dinámico de Estructuras de Obra Civil), 39005 Santander, Spain; 4Smart Engineering, Jordi Girona 1-3 K2M 202c, 08034 Barcelona, Spain; 5Department of Civil and Environmental Engineering, Universitat Politècnica de Catalunya Barcelona Tech, Jordi Girona 1-3, C1, 08034 Barcelona, Spain

**Keywords:** fibre-reinforced concrete, fibre orientation, fibre distribution, inductive method, fibre content

## Abstract

The use of fibre-reinforced concrete (FRC) in structural applications is increasing significantly as a result of (1) the acceptance of this composite into design guidelines and (2) the improvement in terms of sustainability performance that has been reported for cases where FRC has been used. In this context, fibre orientation and distribution are factors that govern the post-cracking response of the FRC. Researchers have already dealt with the analysis of both variables from an experimental and numerical perspective, and design-oriented recommendations were included in existing design guidelines (i.e., *fib* Model Code 2020). Nonetheless, there are still technical aspects to be answered within a research framework before the influence of these variables on the mechanical response of FRC could be covered with sufficient reliability. In this regard, this research is aimed at shedding light on the influence of the mould geometry and concrete pouring/vibration procedures on the fibre orientation and distribution variables as well as on the post-cracking performance of the FRC. An extensive experimental programme aimed at characterising these variables using novel testing techniques (i.e., an inductive non-destructive approach for quantifying fibre amount and orientation and the BCN test for assessing the pre- and post-cracking responses of the FRC) was carried out for this purpose. A relationship has been found between the shape of the formwork and the direction of pouring, along with the direction and distribution of the fibres, both of which proved to have an influence on the residual tensile strength of the concrete. However, it has been confirmed that the first crack resistance depends on the concrete matrix, with the addition of fibres having no relevant influence on that mechanical parameter. The results and conclusions derived from this experimental programme can be extended to FRCs and boundary conditions similar to those established herein.

## 1. Introduction

Fibre-reinforced concrete (FRC) is becoming increasingly important as research progresses, and its use in structural components is being accepted in national and international design guidelines and recommendations [1]. Structural macrofibres (steel or synthetic) are used as unique reinforcement in some applications (i.e., pavements, tunnels, and column-supported flat slabs for buildings, among others) or in combination with conventional reinforcement [2,3,4,5,6,7,8,9].

The addition of fibres improves several mechanical properties (i.e., post-cracking tensile strength and confinement level in compression) and non-mechanical properties (i.e., reduces spalling due to fire loads) of the concrete depending on the type and amount of fibres used [1,8,9,10]. The most significant property of FRC, due to its structural and design implications, is the ability to develop post-cracking flexural/tensile capacity [10], which allows controlling both the propagation and opening of cracks [10,11,12]. Other studies report that the addition of fibres increases Young’s modulus, especially at an early age [13], and improves fatigue behaviour [1,14,15,16,17,18,19]. In addition to the mechanical and geometrical properties of the fibres, it has been demonstrated that the orientation and distribution of the fibres within the matrix have a major impact on the properties of the reinforced concrete. In this regard, the type and amount of fibres influence the fracture energy, which increases as fibre dosage increases [15]. On the other hand, experimental results allow stating that there is a clear relationship between the orientation of the fibres and the mechanical, electrical, and thermal properties of the composite materials [20,21,22,23].

The modulus of elasticity of reinforced concrete depends to a large extent on the orientation of the fibres; the more the fibres are aligned with the direction of stresses, the lower the tensile elastic limit [24]. The residual strength (strength once cracking occurs), which governs the bending strength capacity, also depends on the angle of inclination of the fibres, so that the smaller the angle, the higher the bending strength [25]. Preferential fibre orientation creates weaker planes, favouring increased crack opening at lower load levels [26]. In addition, fibre orientation, in combination with fibre geometry, has an effect on thermal conductivity [27].

It has been shown that the change in fibre orientation and fibre distribution is highly dependent on the length and height of the formwork [24]. Studies conclude that fibres align as a result of the formwork wall effect [7]. The fibre distribution also has a significant influence on the ultimate flexural strength of the FRC, which increases with the increase in the number of fibres per unit area and the dispersion coefficient [25]. P. Martinelli et al. concluded that specimens in contact with the formwork have higher post-cracking stress than those closer to the surface; this is influenced by fibre content (related to segregation) and fibre orientation [28].

Regarding the influence of the rheology of the fresh concrete on the fibre orientation, some authors affirm that the fluidity of the fresh concrete is the governing parameter [23]. This fluidity can be influenced by fibre typology [29] as well as the method and time of concrete vibration [30], among others. These vibratory characteristics are conditioning factors for the positioning and orientation of the fibres. Hence, the performance of the fibres in the residual strength capacity can be altered according to the vibration parameters.

In some studies, equations have been developed to help predict fibre orientation in concrete, as well as CFD-based methods to monitor fibre orientation and distribution [7,24,31,32].

Knowledge of the variation in fibre orientation and distribution is of great importance for improving the mechanical properties of reinforced concrete. Some works, such as [33,34,35], focus on quantifying the effect of concrete flowability/workability on fibre distribution and orientation. Since most of the existing works are based on empirical/analytical models, the need arises for this study, which aims to quantify the orientation/distribution based on non-destructive methods and within the reach of control laboratories to estimate fibre distribution and orientation in quality/production control.

In this paper, a complete characterization of fibre distribution on concrete-reinforced specimens with commercial hooked-end fibres HE++90 or HE++60 is performed depending on the pouring method. The characterisation of this reinforced concrete is a continuation of other studies performed by some of the authors [14,36].

## 2. Materials and Methods

### 2.1. Materials

The concrete was manufactured using a CEM I 52.5N, according to EN 197-1:2011 [37], with a density of 3.12 g/cm^3^, obtained according to UNE 80103:2013 [38]. A superplasticiser additive, MasterEase 5025, was added at 1 wt. % of cement to obtain the target fluidity of the concrete (17 cm of flowability measured using the Abrams cone method).

Hooked-end type steel fibres were selected to manufacture the reinforced concrete. These fibres were HE 90/60 made by ArcelorMittal, and their properties are shown in Table 1 and in Figure 1, where an image of these fibres can be seen.

Limestone gravel and limestone sand were used to produce the concrete. Its physical properties can be seen in Table 2, and Figure 2 shows the grading of the aggregates obtained according to EN 933-1 [39].

The concrete mix proportions were obtained by the Fuller method, adding a 0.44% by volume of steel fibres, and are shown in Table 3.

### 2.2. Methods

#### 2.2.1. Conventional Mechanical Properties

Three different cubic specimens of 150 × 150 × 150 mm^3^ were used to determine compressive strength at 28 days, according to EN 83507:2004 [40]. These specimens were vibrated with an internal vibrator, according EN 12390-2:2020 [41] and cured in a humidity chamber under controlled conditions (20 ± 2 °C and 95 ± 5% humidity) until test moment.

Three different prismatic specimens of 600 × 150 × 150 mm^3^ were used to determine the flexural tensile strength at 28 days, according to UNE-EN 14651:2007 [42]. These specimens were cured in a humidity chamber with the cubes and vibrated in the same way as the cubic specimens. The vertical displacement (δ) was recorded and used to calculate the Crack Mouth Opening Displacement (CMOD), according to the standard, using Equation (1):(1)$$\mathrm{CMOD}=\frac{\delta -0.04}{0.85}$$

Figure 3 shows a specimen being tested to determine the flexural tensile strength.

#### 2.2.2. Pouring Method

To quantify the fibre distribution and its influence on the behaviour of the concrete depending on the direction of pouring, two blocks (initially without holes) of dimensions 1200 × 400 × 1200 mm were manufactured to simulate real dimensions. These blocks were arranged in two different ways, as shown in Figure 4, and the concrete was vibrated with an internal vibrator, according to EN 12390-2:2020 [41].

In the first one, called “BH”, the concrete was poured on the larger side of the cube (1200 × 1200 mm), while the one called “BV” was poured with concrete on one of the smaller sides (1200 × 400 mm).

#### 2.2.3. Specimen Production and Nomenclature

From the previously described concrete blocks, cylindrical specimens with a diameter of 132 mm were extracted using a diamond drilling machine and cut to obtain 132 mm high specimens (L/d = 1). These specimens were coded as follows:A letter and a number that indicate the position on the block (Figure 5);A letter V or H, depending on the block from which they have been extracted;A letter I or S, in the case of the BH block, which indicates whether the tested specimen comes from the top (pouring side) or the bottom (base) of the block.

#### 2.2.4. Inductive Characterisation

The inductive test is an application of Faraday’s law of magnetism and takes advantage of the inductance variations produced by metallic elements when interacting with the magnetic field. For such a test, the specimens must be placed within a plastic container with an established cylindrical geometry. Copper or aluminium wire coils are placed around the container, constituting the sensor element of the system, which is shown in Figure 6. An electrical current goes through the coil and produces a magnetic field around the device, interacting with the steel fibres inside the concrete.

The inductance variation measured when conducting the inductive test is essentially produced by the interaction between the fibres present in SFRC and the magnetic field. In this line, the results of the inductive test mainly depend on the type of fibre (the content of steel in the fibres), as well as the amount and the orientation [8,32,43,44]. Additional parameters such as the type of concrete, the content of water, or the age of the concrete do not influence the magnetic permeability given that the influence of steel on the magnetic field is several orders of magnitude higher than that of concrete. Accordingly, it is possible to simplify the inductance measurements by disregarding the contribution of concrete to the inductance variation. The physical parameter of the magnetic field induced is the magnetic inductance (in mmH), whose magnitude is modified by the presence (type and orientation) of steel fibres.

Calibration must be done beforehand using samples with variable fibre content that need to be characterised by means of the inductive test and subsequently crushed and ground to weigh the fibres inside the sample. The calibration curve is calculated using a zero value of inductance for a sample with no fibre content and the relationship between inductance and the weight of fibres in the analysed and crushed samples. Once this calibration is done, no further calibration is required if the same type of fibre is used.

The test can be conducted in both cubic and cylindrical specimens [32] since these geometries allow the characterization of both moulded and core-drilled test samples. Actually, it is possible to determine the fibre content with an error below 3% and with a low scatter on the measurements.

#### 2.2.5. Barcelona Test

The Barcelona test (BCN) responds to a double punch test configuration applied to a cylindrical (300 × Ф150 mm) or cubic (150 mm) specimen. It is carried out by applying a vertical force above the specimen through cylindrical steel punches of 25 mm height, the diameter of ¼ of the cross-section minor dimension. The load is applied at 0.5 ± 0.05 mm/min through the upper punch. During the loading process, a triaxial compressive stress state occurs at both extreme faces, and internal cones are generated (one per face). As the cones penetrate inside the specimen, vertical cracks that open circumferentially appear, and fibres prevent these cracks from widening through a residual tensile concrete strength mechanism. A Load-Total Circumferential Opening Displacement (TCOD) curve or Load-Axial Displacement (vertical) can be derived from the test (Figure 7).

The BCN was originally developed to control the TCOD through a circumferential extensometer, as described in the standard UNE 83515. The test procedure was simplified through the definition of a correlation between the TCOD and the vertical displacement [32,45,46] to use the latter as the control variable. The analytical and theoretical description of such a correlation is described in Figure 8, which shows how the load-axial displacement curve can be used to show the results of load-TCOD.

The correlation between the circumferential opening and the vertical displacement allows replacing cylindrical specimens with cubic samples. Even though the use of cylindrical specimens for testing is an advantage in cases of extracting cores from existing structures, testing cubic specimens under the BCN and properly analysing the results could provide an evaluation of the fibres’ preferential orientation [45].

Cracks should propagate in the direction corresponding to the preferential orientation of the fibres, meaning that the minimum orientation direction is perpendicular to the crack. For this, cubic specimens can be tested in different directions to take advantage of the relationship between post-cracking strength and fibre orientation. This provides an estimation of the orientation of fibres in % according to the contribution of the fibres in each direction to the residual strength [45].

#### 2.2.6. Data Curation

##### Statistical Analysis

Several statistical analyses were carried out using the SciPy Python library:T-Test: It is an analysis that allows the comparison of the mean value of two data distributions. In general, if a *p*-value of less than or equal to 0.05 is obtained, it is assumed that there is evidence of significant differences between the means of the distributions;Levene’s Test: It is an analysis that allows the comparison of the standard deviation of two data distributions. In general, if a *p*-value of less than or equal to 0.05 is obtained, it is assumed that there is evidence of significant differences between the standard deviation of the distributions;Shapiro–Wilk: This is a test used to check the normality of a data set. In general, the test rejects the hypothesis of normality when the *p*-value is less than or equal to 0.05.

##### Correlation Matrix

The Python libraries Pandas and Matplotlib were used to generate the correlation matrix. The Pearson correlation coefficient (r) is visually represented in a correlation matrix. That is, the squared values will be the coefficient of determination (r^2^), and values near -1 indicate a strong inversely proportional correlation. Values close to 1 indicate a high correlation in a directly proportional way, and those values close to 0 will imply a low correlation between the variables.

## 3. Results and Discussion

### 3.1. Mechanical Properties

The mean compressive strength resulting from three specimens was *f*_cm_ = 50 MPa (CoV = 7.4%). The mean residual flexural strength for the 0.5 mm crack opening (*f*_Rm,1_, CMOD = 0.5 mm) was 3.5 MPa (CoV = 7.1%), and for the 2.5 mm crack opening (*f*_Rm,3_, CMOD = 2.5 mm), it was 6.3 MPa (CoV = 8.3%). In Figure 9, a compressive and a flexural tested specimen can be seen.

### 3.2. Inductive Tests

Figure 10 depicts the distribution of the amount of fibre as a function of the concrete pouring method (a) and as a function of height for BH (b). Table 4 shows the results of the *t*-test and the Levene’s test, which allow a comparison of the two distributions, and gathers the mean value, standard deviation, and *p*-value of a *t*-test in the case of BH.

It is concluded that the amount of fibre is equivalent in both types of pouring, but that the results of vertical pouring allow proving that the distribution of the amount of fibre is more homogeneous than in horizontal concreting (note the differences in CoVs). Thus, according to the results in Table 4, in BV (vertical concreting), the mean fibre distribution coincides with the theoretical distribution (35 kg/m^3^) and with a significantly lower dispersion than that obtained for BH. Longer concrete face travel results in better fibre distribution, which is consistent with other studies [33].

Concerning the effect of the height of the cross-section poured (BH top and bottom), the *t*-test provides a P_value_ of 4.6 × 10^−7^, allowing the statement that, with high probability, the height has a significant influence on the amount of fibre distribution. In this regard, the average fibre amount at the top (24.2 kg/m^3^) and bottom (39.7 kg/m^3^) of the BH prism is 31% and 13% lower and greater, respectively, in comparison to the theoretical fibre amount (35 kg/m^3^). This is due to a slight segregation effect [47,48], with the fibres being denser than the concrete itself, and also due to the potential energy, which is bigger in BV because the flow is higher.

Once it had been verified that in the case of BH there were significant differences depending on the position of the cores, it was then analysed to see if these variations were also observed in the case of BV. Table 5 shows the results of the mean value (each obtained from 12 values), standard deviation, and Shapiro–Wilk’s *p*-value according to the row and column for which it is computed, from which it can be concluded that in all cases the available data conform to a normal distribution.

It can be seen from the results obtained that there is more dispersion where there are more fibres (more than 10% in values higher than the average of 35 kg/m^3^ of fibres).

Table 6 presents the mean value and standard deviation values as a function of the exact position of the sample (for a given row and column condition). The results of hypothesis testing are shown in Table 7. The *t*-test results carried out on the sample means (of each zone) reveal the differences of the means are not significant and, with high probability, these belong to the same sample population. Consequently, it can be stated that position is not an influencing variable for the fibre amount distribution.

According to the results obtained and the statistical tests performed, it can be concluded that the fibre amount distribution tends to be more uniform in the case of pouring the slabs from the vertical direction. According to the results obtained, the most suitable pouring method to achieve better homogeneity according to the number of fibres distributed in the slab would be the BV.

Figure 11 shows the fibre distribution in each plane for each slab, differentiating the top and bottom of the BH.

It can be observed that in all cases, the largest number of fibres (more than 36%) are oriented in the XY plane, which means that they are mostly aligned with the larger face of the formwork, both in the horizontal and vertical pouring directions.

Focusing on the case of the horizontal concreting direction, it can be observed that in the upper part of the block this tendency is less clear, which may be due to the fact that the fibres have less time and space to align with the flow.

### 3.3. Barcelona Test

Regarding the influence of the pouring procedure on the Barcelona test results (pre- and residual strengths), Figure 12 gathers the values of cracking (F_ct_) and residual strengths (R_s_) as a function of the type of concreting (a) and of the height in BH (b). The residual strengths (R_s_) are given for crack openings of 0.5, 1.5, 2.5, and 3.5 mm. Table 8 shows the statistical results for these parameters.

Based on the results obtained, the method of pouring does not influence the cracking load, but it does influence the residual properties of the concrete. This is due to the influence of the concreting method on the fibre orientation, an influence that other studies [33,35] have affirmed. In all cases, as in the case of BV, lower residual strength values are obtained.

Based on the results obtained in BH, it can be concluded that the position in height does have a certain influence on the post-cracking strength properties of the concrete, which makes sense considering that it has been found that there is a higher amount of fibres in the lower part of the concrete.

From Figure 11, it can be seen that for large deformation values, there is a very precise correlation between values, whereas for small values, the distortion increases markedly. This is due to the fact that for larger deformations, the fibres are the elements that are resisting the force almost completely, which, being metallic, have a more linear behaviour. At smaller deflections (crack openings), the fibre has more influence on the cracking load, as there is more fibre surface “gripping” both parts of the concrete.

### 3.4. Correlation between the Inductive Test and the BCN Test

Figure 13 shows the correlation matrix corresponding to the results of the inductive test (which measures the amount and orientation of the fibres) and the Barcelona test.

From Figure 13 it can be seen firstly that the parameter F_ct_ is not highly related to any other of the variables analysed. This is because it is a parameter depending on the concrete matrix, and throughout this study, the concrete (without fibres) has been the same in all cases.

From Figure 13, it can be seen that the circumferential deformation values are highly correlated with each other, especially in the case of Rs_2.5_ and Rs_3.5_. In particular, it can be seen that this correlation is higher as the circumferential deformation value increases. Figure 11 shows the circumferential deformation values compared as a function of the deformation value.

Regarding the correlation between the results of the inductive test, a high linear correlation between all parameters is observed. Regarding the correlation between residual strength values and fibre quantity, it can be seen that the correlation is rather small, but that it correlates better with the projected fibre values in the XY plane than with total fibre values. This is because fibres oriented perpendicular to the direction of the force provide the most resistance.

In Figure 14, it can be seen the amount of fibre in XY vs. circumferential deformations in BH, which differentiate between the top and bottom of the slab.

The circumferential deformation depends on the amount of fibre in the XY plane (parallel to the ground), so that the smaller the crack opening, the more dependent it is. Furthermore, it can be seen from Figure 14 that the amount of fibre in this plane is higher in the lower part of the block, which results in higher resistances. Furthermore, the results are more stable with smaller crack opening (higher correlation coefficient), so as the crack opening increases, the results become more heterogeneous. This is because the smaller the crack opening, the more fibre surface is in contact with the matrix on both sides of the crack.

This means that the fibres are oriented parallel to the direction of flow, so that they flow from the centre (where the concrete is poured) to the sides. The fibres are oriented preferentially in the direction of the larger faces of the formwork.

In Figure 15, the correlation between circumferential deformation for a 0.5 mm crack opening and the amount of fibre in the XY plane can be seen.

There is a trend between the number of fibres in the XY plane and the circumferential deformation in the BH. However, in the BV, both parameters are not correlated.

Referring to the different parts of the BH, Figure 16 represents the quantity of fibres with respect to the circumferential deformation for a crack opening of 0.5 mm.

Figure 17 shows the correlation between circumferential deformation and fibre quantity in the XY plane for both types of pouring directions.

As shown in Figure 18, the lower part of the slab has a higher number of fibres. Furthermore, it can be seen that it is in this lower part where the greatest deformations are achieved.

## 4. Conclusions

For the specific case study of reinforced and internally vibrated concrete with a fibre dosage of 35 kg/m^3^, the main conclusion drawn from the analysis of the results is that the method of pouring has little influence on the cracking load and a great influence on the behaviour of the cracked concrete, with the following conclusions in particular:In horizontal pouring (BH), the preferential fibre direction is in the plane parallel to the bearing face on the ground (a mean of 38.18% of fibres are oriented in the XY plane), while in vertical concreting, the preferred orientation is the face perpendicular to the ground. There is an influence of the formwork walls on the orientation of the fibres in the concrete, so that the fibres tend to align with the larger faces;The pouring method does not influence the cracking load (Fct) but does influence the residual strengths, which are higher the more fibres are perpendicular to the load (XY plane);In a vertical pour, homogeneity according to the number of fibres distributed in the slab is achieved;There is a correlation between circumferential deformations, which is more accurate at high crack opening values;In the lower part of the BH, there are more fibres than in the upper part (approximately twice the density), which in turn influences the residual strength, which is higher in the lower part;According to the results of the experimental program, both the BCN and inductive tests are appropriate for quantifying the amount of fibres within the concrete, their orientation, and the post-cracking indirect strength.

## Figures and Tables

**Figure 1 materials-16-01404-f001:**
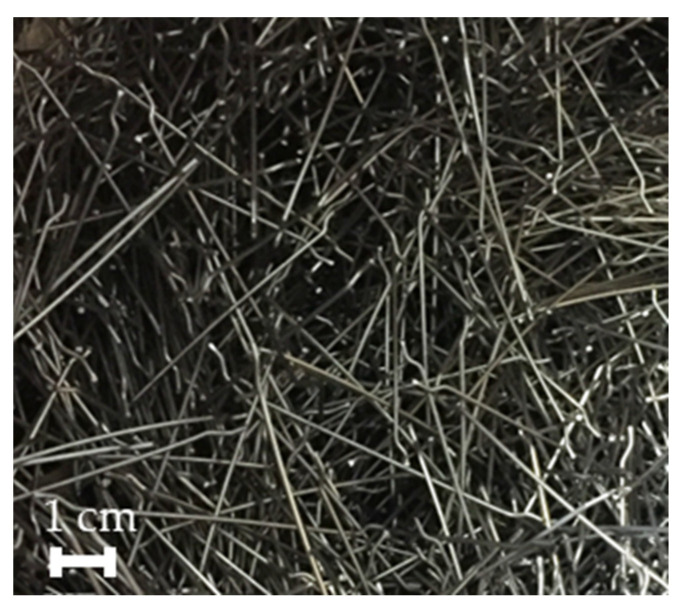
Hooked-end steel fibres used.

**Figure 2 materials-16-01404-f002:**
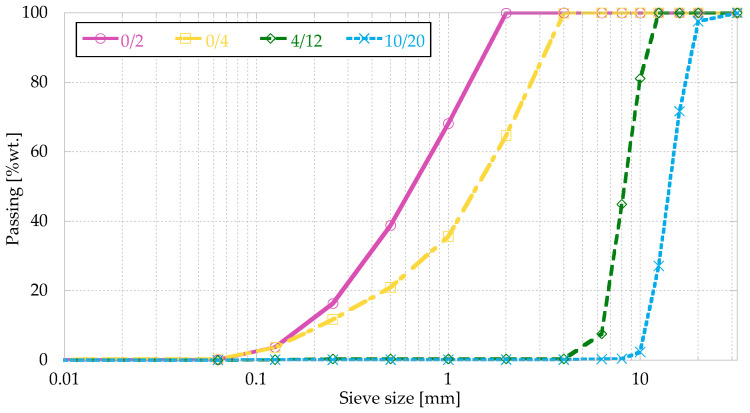
Grading curve.

**Figure 3 materials-16-01404-f003:**
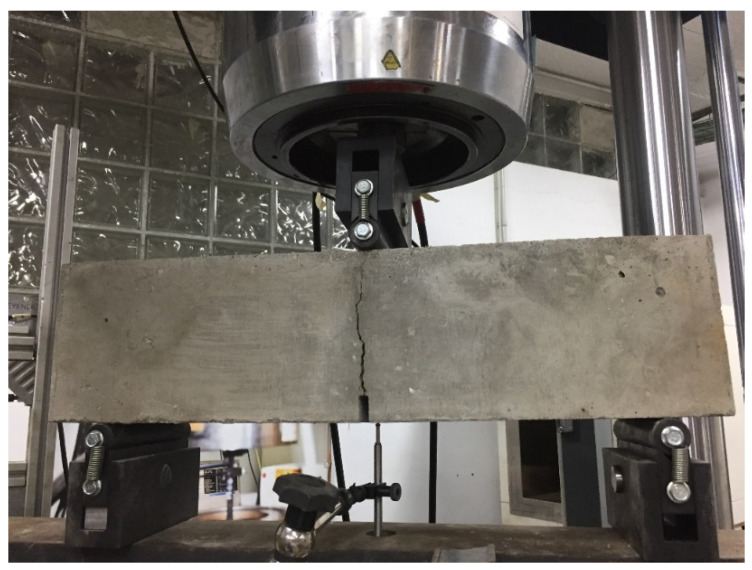
Flexural tensile strength test of the fibre-reinforced concrete.

**Figure 4 materials-16-01404-f004:**
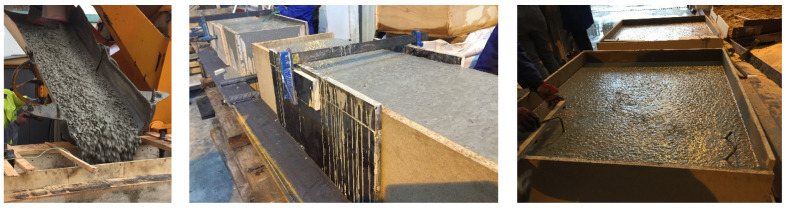
Concrete pouring (**left**), BV (**centre**), and BH (**right**).

**Figure 5 materials-16-01404-f005:**
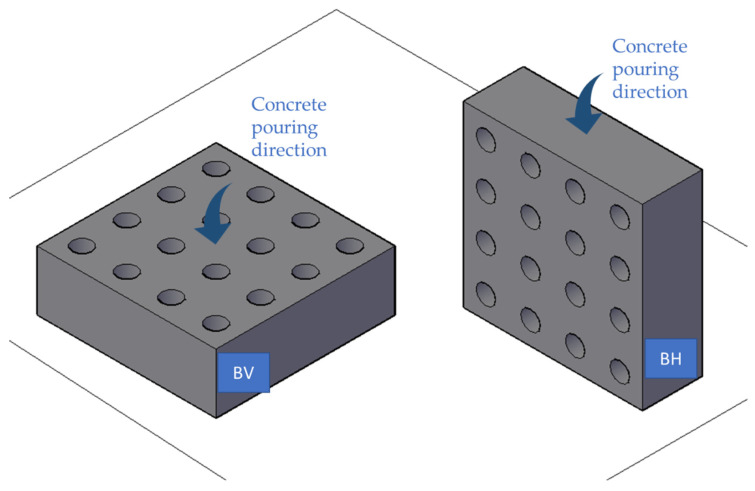
Concrete pouring directions in each block.

**Figure 6 materials-16-01404-f006:**
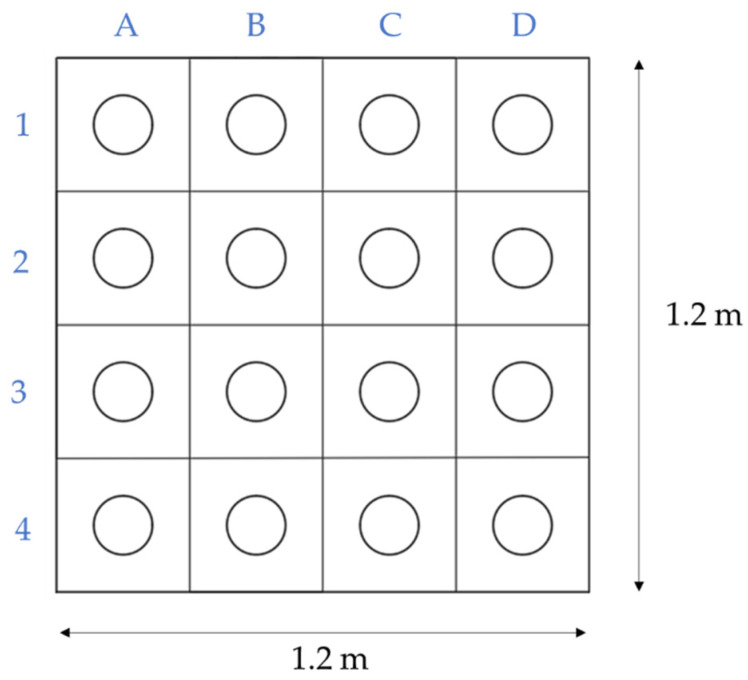
Code of the extracted specimens.

**Figure 7 materials-16-01404-f007:**
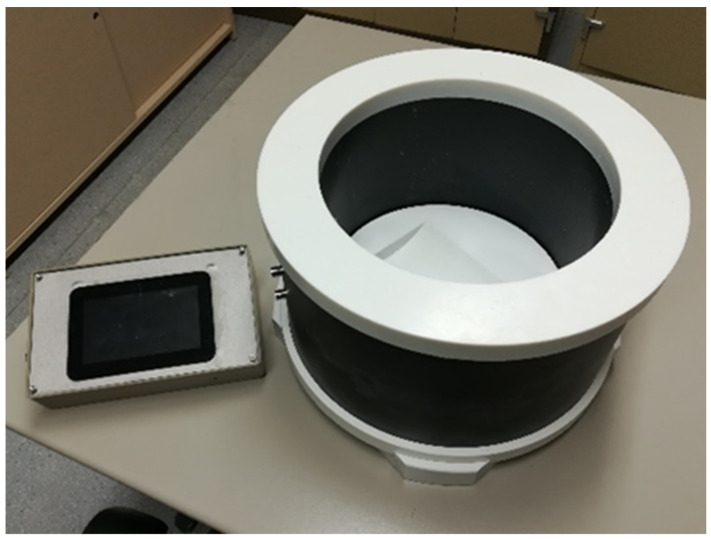
SmartFibreC® inductive equipment for measuring the amount and orientation of steel fibres in hardened concrete.

**Figure 8 materials-16-01404-f008:**
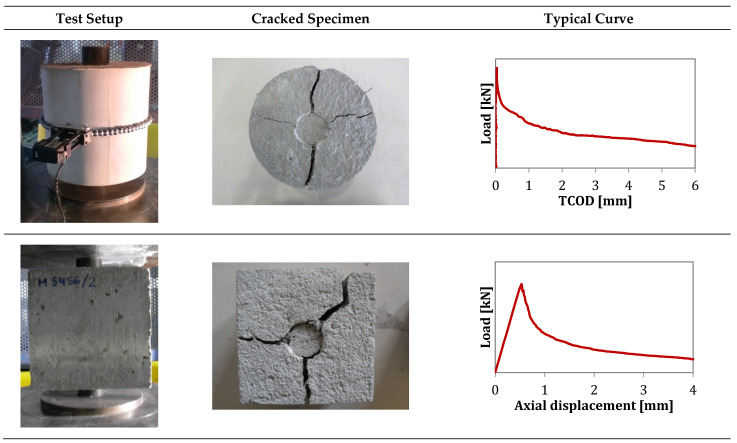
Barcelona test in cylindrical and cubic specimens.

**Figure 9 materials-16-01404-f009:**
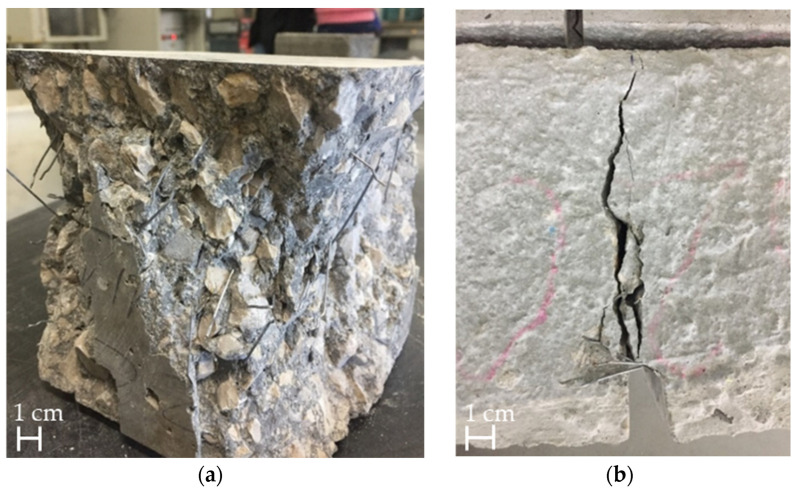
Specimens after compressive test (**a**) and flexural test (**b**).

**Figure 10 materials-16-01404-f010:**
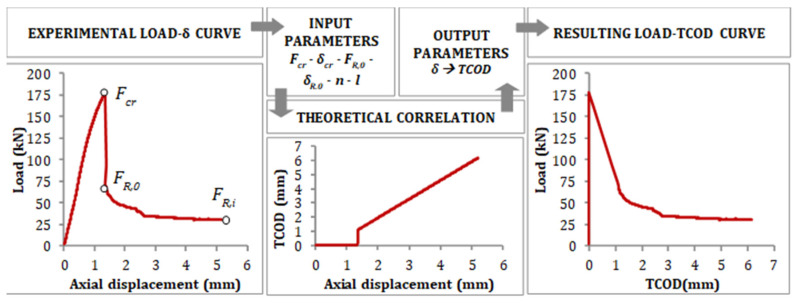
Correlation between axial displacement and TCOD.

**Figure 11 materials-16-01404-f011:**
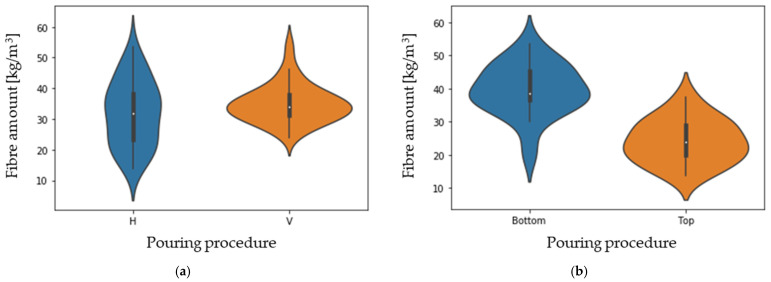
Fibre amount distributions as functions of the pouring procedure (**a**) and of the height for BH (**b**).

**Figure 12 materials-16-01404-f012:**
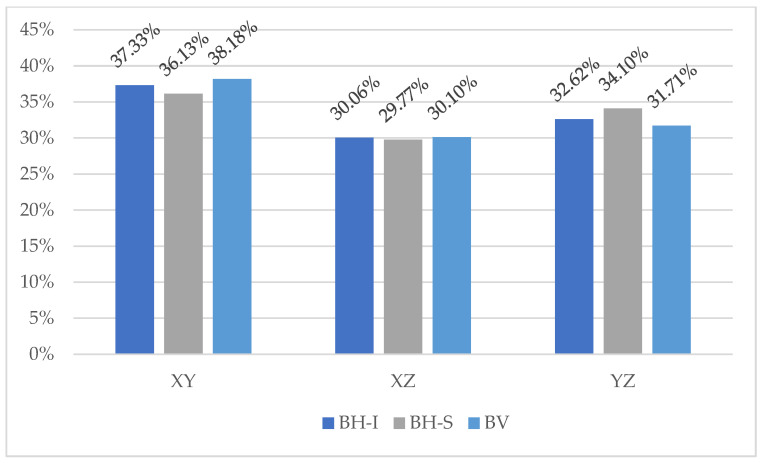
Distribution of fibres in each plane.

**Figure 13 materials-16-01404-f013:**
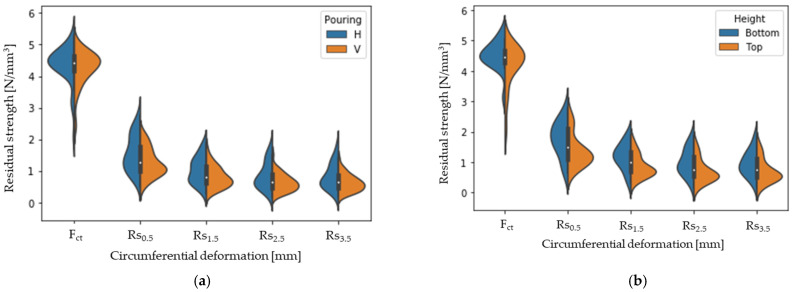
Distribution of the residual strength obtained by the BCN test as a function of the pouring procedure (**a**) and as a function of the height in BH (**b**).

**Figure 14 materials-16-01404-f014:**
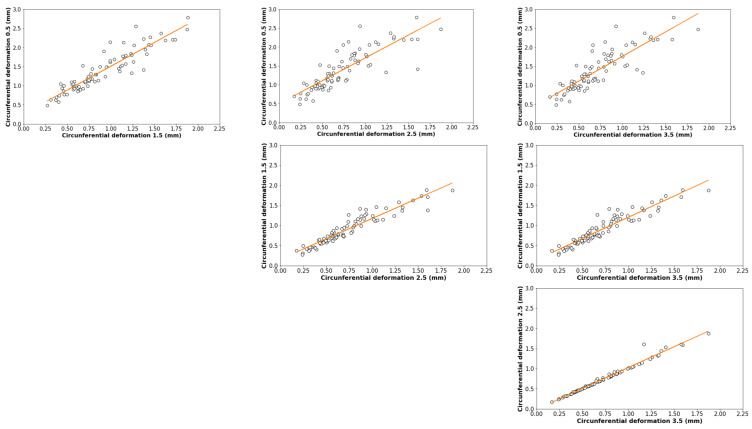
Correlation between different values of circumferential deformation.

**Figure 15 materials-16-01404-f015:**
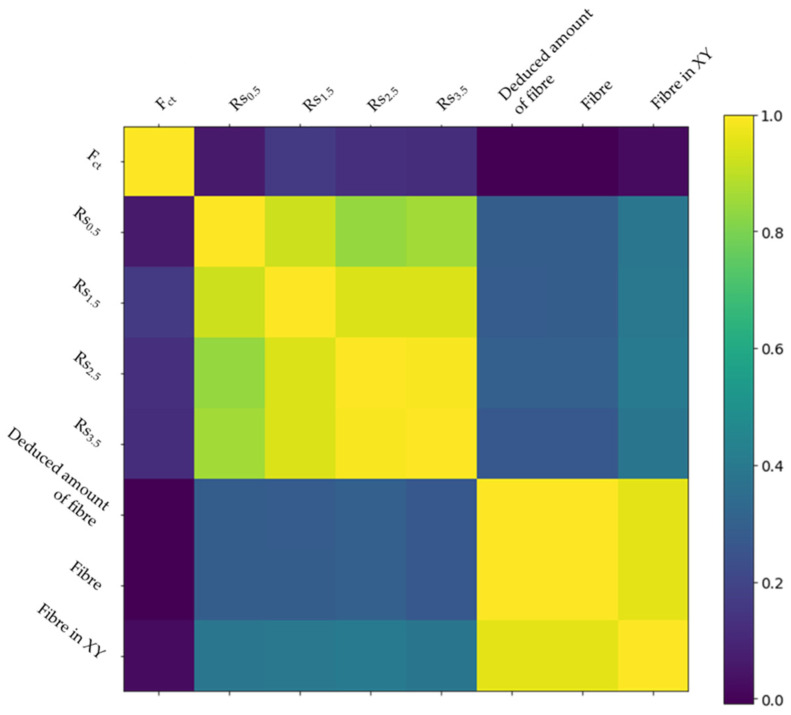
Correlation plot.

**Figure 16 materials-16-01404-f016:**
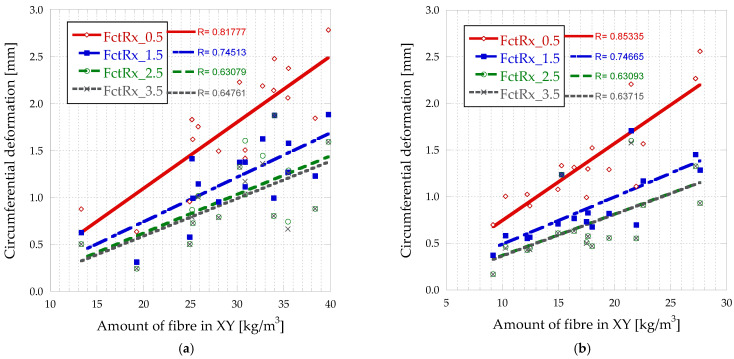
Amount of fibre in XY vs. circumferential deformations in BH-bottom (**a**) and BH-top (**b**).

**Figure 17 materials-16-01404-f017:**
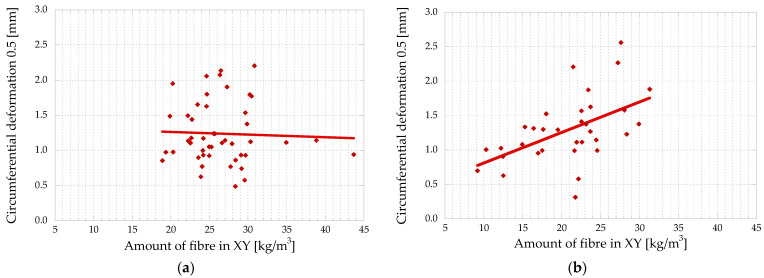
Correlation between circumferential deformation vs. fibre quantity in the XY plane for BV (**a**) and BH (**b**).

**Figure 18 materials-16-01404-f018:**
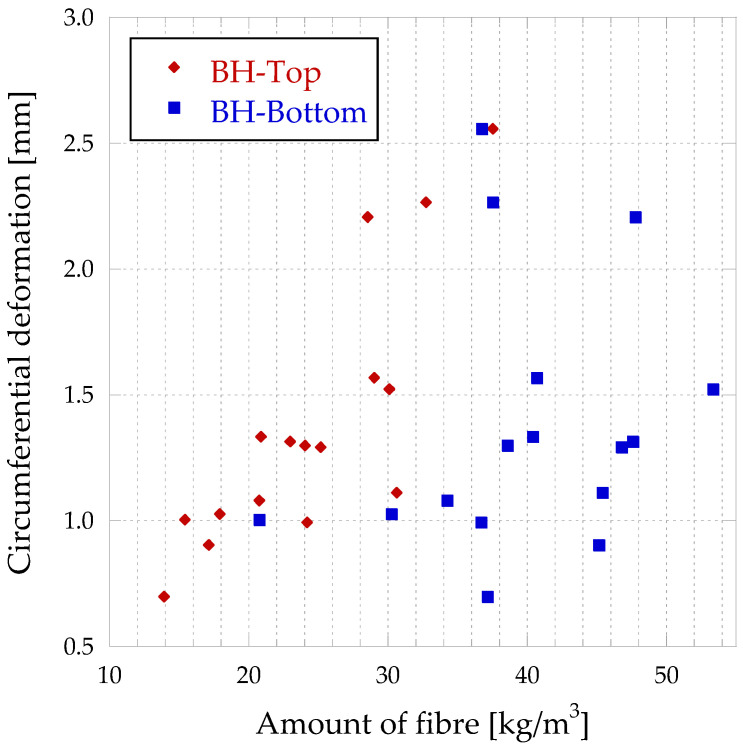
Correlation between circumferential deformation vs. fibre quantity for BH as a function of the height position.

**Table 1 materials-16-01404-t001:** Steel fibre properties.

Parameter	Value
Fibre shape	Hooked end
Length (mm)	60
Diameter (mm)	0.90
Aspect ratio (L/d)	67
Tensile strength (MPa)	1200

**Table 2 materials-16-01404-t002:** Aggregate physical properties.

Size [mm]	Sand Equivalent	Absorption [%]	Density [g/cm^3^]
0/2	>75	0.49	2.69
0/4	>80	0.49	2.69
4/12	-	0.54	2.70
10/20	-	0.54	2.68

**Table 3 materials-16-01404-t003:** Mix proportions.

Material	Mix [kg/m^3^]
Cement	390
0/4	480
4/12	480
10/20	480
Water	165
Additive	3.9 (1 wt. % of cement)
Fibres	35

**Table 4 materials-16-01404-t004:** Statistical parameters that influence the pouring procedure’s influence on the fibre amount distribution.

		Mean	STD (CoV)	P_value_ (T-Test)	P_value_ (Levene’s Test)
Influence of the pouring procedure	BH	31.9	10.6 (33.2%)	1.0 × 10^−1^	1.7 × 10^−4^
	BV	35.0	6.3 (18.0%)		
Influence of height in BH	BH-Top	24.2	6.5 (26.9%)	4.6 × 10^−7^	-
	BH-Bottom	39.7	7.7 (19.4%)		

**Table 5 materials-16-01404-t005:** Statistical description of the fibre quantity depending on the position.

Position	Mean	STD (CoV)	Confidence Interval 95%	Shapiro–Wilk (P_value_)
Row 1	36.0	8.3 (23.1%)	36.0 ± 5.3	0.45
Row 2	36.1	7.4 (20.5%)	36.1 ± 4.7	1.00
Row 3	34.6	3.4 (9.8%)	34.6 ± 2.2	0.98
Row 4	32.1	3.0 (9.3%)	32.1 ± 1.9	0.54
Column A	31.5	4.3 (13.7%)	31.5 ± 2.7	0.89
Column B	37.4	4.8 (12.8%)	37.4 ± 3.0	0.94
Column C	34.5	3.5 (10.1%)	34.5 ± 2.2	0.36
Column D	35.8	9.4 (26.3%)	35.8 ± 6.0	0.07

**Table 6 materials-16-01404-t006:** Statistical description of the fibre quantity depending on the position.

	Mean Value [kg/m^3^]				Standard Deviation [kg/m^3^]				CoV			
	A	B	C	D	A	B	C	D	A	B	C	D
1	30.57	39.13	34.45	40.07	3.86	6.84	3.58	14.72	12.6%	17.5%	10.4%	36.7%
2	28.52	40.30	37.77	37.98	4.24	4.55	1.82	11.74	14.9%	11.3%	4.8%	30.9%
3	36.04	36.19	35.23	30.77	4.29	2.46	2.16	2.53	11.9%	6.8%	6.1%	8.2%
4	30.87	34.04	30.49	33.67	2.32	4.29	2.48	0.92	7.5%	12.6%	8.1%	2.7%

**Table 7 materials-16-01404-t007:** Analysis of significant differences between positions according to the *t*-test.

	Columns						Rows					
	A–B	A–C	A–D	B–C	B–D	C–D	1–2	1–3	1–4	2–3	2–4	3–4
P_value_	0.005	0.077	0.166	0.103	0.103	0.655	0.978	0.569	0.154	0.508	0.110	0.088

**Table 8 materials-16-01404-t008:** P_value_ of a *t*-test on the effect of pouring procedure on concrete resistance properties.

	F_ct_	Rs_0.5_	Rs_1.5_	Rs_2.5_	Rs_3.5_
**BH mean value**	4.35	1.57	1.04	0.86	0.84
**BH std value**	0.62	0.58	0.43	0.43	0.41
**BV mean value**	4.26	1.26	0.81	0.64	0.63
**BV std value**	0.55	0.44	0.33	0.28	0.26
**P_value_ (T-Test)**	0.48	7 × 10^−3^	8 × 10^−3^	5 × 10^−3^	6 × 10^−3^
**Top mean value**	4.19	1.37	0.88	0.71	0.71
**Top std value**	0.71	0.51	0.36	0.37	0.37
**Bottom mean value**	4.51	1.77	1.19	1.01	0.97
**Bottom std value**	0.47	0.59	0.43	0.45	0.42
**P_value_ (T-Test) in BH**	0.14	0.04	0.03	0.04	0.06

## Data Availability

Not applicable.

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
