# Peer review of "Statistical Analysis of the Pouring Method’s Influence on the Distribution of Metallic Macrofibres into Vibrated Concrete"

_materials, 2023, doi:10.3390/ma16041404_

Round 1

Reviewer 1 Report

- Although the title of the paper is "Influence of pouring method on the distribution of metallic macrofibres into vibrated concrete" you did not specify if the compaction of specimens were in the moulds using hand tamping, vibrating table, or internal vibrator as it specified in EN 12390-2 or as is written in EN 14651 "Compaction shall be carried out by external vibration";

- in my opinion, the samples reinforced with short fibers should be compared in the BV/BH positions with unreinforced samples in the BV/BH positions;

-  please correct the reference 40, est=Test;

Author Response

Responses to the reviewers’ comments to the manuscript materials-2080494

Influence of pouring method on the distribution of metallic macrofibres into vibrated concrete

----

Dear Reviewers,

We greatly appreciate the opportunity you give us to improve the paper with your valuable comments. Here you can find the detailed comments and performed changes.

Best regards,

The authors

Responses to comments:

Reviewer #1 (Changes in red)

Although the title of the paper is "Influence of pouring method on the distribution of metallic macrofibres into vibrated concrete" you did not specify if the compaction of specimens were in the moulds using hand tamping, vibrating table, or internal vibrator as it specified in EN 12390-2 or as is written in EN 14651 "Compaction shall be carried out by external vibration";

Thanks for the comment. The sentence “and the concrete was vibrated with internal vibrator, according EN 12390-2:2020” was added.

In my opinion, the samples reinforced with short fibers should be compared in the BV/BH positions with unreinforced samples in the BV/BH positions;

Thank you very much for your suggestion, we will be able to take this into account in future works since we have not made slabs without fibers of similar dimensions for this study.

Please correct the reference 40, est=Test;

Many thanks for the observation, the reference was corrected.

Reviewer 2 Report

Please refer the attachment

Author Response

Responses to the reviewers’ comments to the manuscript materials-2080494

Influence of pouring method on the distribution of metallic macrofibres into vibrated concrete

----

Dear Reviewers,

We greatly appreciate the opportunity you give us to improve the paper with your valuable comments. Here you can find the detailed comments and performed changes.

Best regards,

The authors

Responses to comments:

Reviewer #2 (Changes in green)

This is an interesting study and the authors have collected adequate dataset using statistical analysis and correlation matrix. The paper is generally well written and structured. However, in my opinion the paper has some shortcomings in regards to some data analyses and text, and I feel this unique dataset has not been utilized to its full extent. Furthermore I made additional suggestions for more in-depth analyses of the data. Key critical points are the relation of the result from each testing to the next testing. For example, between mechanical testing and inductive testing. The length of paragraph in introduction was not equivalents. Given these shortcomings the manuscript requires major revisions.

Thank you very much for these encouraging comments and for giving us the opportunity to improve our work.

  1. The title is better to have the term “statistical”.

Based on your comment, the following new title is proposed:

“Statistical analysis of pouring method influence on the distribution of metallic macrofibres into vibrated concrete”

  1. There is a duplicate of Figure 1 in page 3.

Could you please provide more details on where the duplicate is located? After reviewing the document, we have not been able to find it.

  1. Please label which measurements of the cylindrical sample stated in 2.2.3 which was 132nm.

The extracted samples were 132 mm in diameter. These samples were then cut so that their height was also 132 mm. Thus, 132 mm corresponds to both diameter and height. To make it clearer, the length-diameter ratio has been added: “l/d=1”.

  1. Remove the word method title 2.2.4

Indeed, section 2.4 should be entitled “Inductive characterization” instead of “Inductive characterization method.”

  1. Paragraph 3-4 in subunit 2.2.4 is more suitable in literature. Please be more specify about the parameter used for inductive method.

We have added: “The physical parameter of the magnetic field induced is the magnetic inductance (in mmH), which magnitude is modified by the presence (type and orientation) of steel fibres”

  1. Please state the details about the compressive and flexural test (eg : types of equipment, standard)

In section 2.2.1 it has been indicated that the standard used for the compression test is EN 83507:2004 and cubic specimens of 150x150x150 mm3, and for the flexural test, the standard UNE-EN 14651:2007 using prismatic specimens of 600x150x150 mm3.

  1. The author only mentions “the largest number of fibres” “fewer fibres” in the paragraphs, please be more specify in the discussions.

Thank you for your appreciation. Some specific value has been added during the article. In the first case, “more than 36%”. In the second case it was a mistake and the sentence was corrected and we have added specific value: “It can be seen from the results obtained that there is more dispersion where there are greater fibres (more than 10% in values higher than the average of 35 kg/m3 of fibres)”.

  1. Please rearrange the graph in Fig 13.

Fig 13. looks like that on purpose. This is because it compares in each row and each column always the same parameter. In this way, in our opinion, it makes the interpretation of the results easier.

  1. The author did mention that “the fibres are oriented parallel to the direction of flow”, is there any micrograph image to support this statement?

This conclusion in based in the numerical results of the inductive method.

Reviewer 3 Report

This paper reported the influence of pouring method on the distribution of metallic macrofibres into vibrated concrete. The research is interesting and the results are promising. They are sufficient to represent as a standalone paper. The paper is prepared at a standard level but before proceeding to publication, below minor revision needs to be addressed.

1.Abstract should be more focused and the results to be presented.

2. Introduction:“Regarding the influence of the rheology of the fresh concrete on the fibre orientation, some authors affirm that the fluidity of the fresh concrete is the governing parameter [29]: The research on the relationship between the rheology of fresh concrete and the distribution and orientation of fiber should be further summarized.

3. 2.2.2 Pouring methodPlease further explain the equipment used for pouring, and it is better to supplement the photos of on-site pouring.

4. If vibration is carried out after the pouring of the test blocks. If yes, please describe the specific molding method and vibration time. In my opinion, vibration method has significant influence on the fiber orientation and distribution. The influence of vibration should be considered in this study, especially when the bottom fiber is significantly more than the top fiber

5. The conclusion needs to be further written.

Author Response

Responses to the reviewers’ comments to the manuscript materials-2080494

Influence of pouring method on the distribution of metallic macrofibres into vibrated concrete

----

Dear Reviewers,

We greatly appreciate the opportunity you give us to improve the paper with your valuable comments. Here you can find the detailed comments and performed changes.

Best regards,

The authors

Responses to comments:

Reviewer #3 (Changes in blue)

This paper reported the influence of pouring method on the distribution of metallic macrofibres into vibrated concrete. The research is interesting and the results are promising. They are sufficient to represent as a standalone paper. The paper is prepared at a standard level but before proceeding to publication, below minor revision needs to be addressed.

1.Abstract should be more focused and the results to be presented.

Thank you for your appreciation. The abstract has been modified by adding this text: “A relationship has been found between the shape of the formwork and the direction of pouring with the direction and distribution of the fibres; both proved to have an influence on the residual tensile strength of the concrete. However, it has been confirmed that the first crack resistance depends on the concrete matrix, the addition of fibres having no relevant influence on that mechanical parameter.”

  1. Introduction:“Regarding the influence of the rheology of the fresh concrete on the fibre orientation, some authors affirm that the fluidity of the fresh concrete is the governing parameter [29]”: The research on the relationship between the rheology of fresh concrete and the distribution and orientation of fiber should be further summarized.

Thank you for your comment. A note on the influence of vibration and type of fibers on the fresh state properties has been added in the abstract as follows: “This fluidity can be influenced by fibre typology [30] and the method and time of vibration of the concrete [31], between others. These vibrating characteristics are conditioning factors for the positioning and orientation of the fibres. Hence, the performance of the fibres in the residual strength capacity can be altered according to the vibration parameters.”

  1. 2.2.2 Pouring method: Please further explain the equipment used for pouring, and it is better to supplement the photos of on-site pouring.

Thanks, these photos have been included in Fig 4:

Fig.4. Concrete pouring (left), BV (centre) and BH (right).

  1. If vibration is carried out after the pouring of the test blocks. If yes, please describe the specific molding method and vibration time. In my opinion, vibration method has significant influence on the fiber orientation and distribution. The influence of vibration should be considered in this study, especially when the bottom fiber is significantly more than the top fiber

We agree that vibration will possibly influence the position of the fibers. However, we have not performed the fresh state tests necessary to make this assertion. This may be the subject of further work by the authors and we appreciate the appreciation.

The vibration method, using an external vibrator, has been added.

  1. The conclusion needs to be further written.

We have added other conclusion: “the results of the experimental program allow stating that both the BCN and inductive tests are suitable to quantify the amount of fibres within the concrete, their orientation and the post-cracking indirect strength”

Round 2
